# Chitosan/Polyvinyl Alcohol/Tea Tree Essential Oil Composite Films for Biomedical Applications

**DOI:** 10.3390/polym13213753

**Published:** 2021-10-29

**Authors:** Jorge Iván Castro, Carlos Humberto Valencia-Llano, Mayra Eliana Valencia Zapata, Yilmar Joan Restrepo, José Herminsul Mina Hernandez, Diana Paola Navia-Porras, Yamid Valencia, Cesar Valencia, Carlos David Grande-Tovar

**Affiliations:** 1Grupo de Investigación SIMERQO, Departamento de Química, Universidad del Valle, Calle 13 No. 100-00, Cali 76001, Colombia; jorge.castro@correounivalle.edu.co; 2Grupo Biomateriales Dentales, Escuela de Odontología, Universidad del Valle, Calle 4B No. 36-00, Cali 76001, Colombia; carlos.humberto.valencia@correounivalle.edu.co; 3Grupo de Materiales Compuestos, Escuela de Ingeniería de Materiales, Facultad de Ingeniería, Universidad del Valle, Calle 13 No. 100-00, Santiago de Cali 760032, Colombia; valencia.mayra@correounivalle.edu.co (M.E.V.Z.); yilmar.restrepo@correounivalle.edu.co (Y.J.R.); jose.mina@correounivalle.edu.co (J.H.M.H.); 4Grupo de Investigación Biotecnología, Facultad de Ingeniería, Universidad de San Buenaventura Cali, Carrera 122 # 6-65, Cali 76001, Colombia; dpnavia@usbcali.edu.co; 5Área de Investigación, Desarrollo e Innovación, Disproquin S.A.S., Calle 93 Número 7u-2a, Vía Cali-Juanchito 760021, Colombia; analista.id@disproquin.com.co (Y.V.); analista.id2@disproquin.com.co (C.V.); 6Grupo de Investigación de Fotoquímica y Fotobiología, Facultad de Ciencias, Universidad del Atlántico, Carrera 30 Número 8-49, Puerto Colombia 081008, Colombia

**Keywords:** biocompatibility, chitosan, tea tree essential oil, membrane, polyvinyl alcohol, tissue engineering

## Abstract

Tissue engineering is crucial, since its early adoption focused on designing biocompatible materials that stimulate cell adhesion and proliferation. In this sense, scaffolds made of biocompatible and resistant materials became the researchers’ focus on biomedical applications. Humans have used essential oils for a long time to take advantage of their antifungal, insecticide, antibacterial, and antioxidant properties. However, the literature demonstrating the use of essential oils for stimulating biocompatibility in new scaffold designs is scarce. For that reason, this work describes the synthesis of four different film composites of chitosan/polyvinyl alcohol/tea tree (*Melaleuca alternifolia*), essential oil (CS/PVA/TTEO), and the subdermal implantations after 90 days in Wistar rats. According to the Young modulus, DSC, TGA, mechanical studies, and thermal studies, there was a reinforcement effect with the addition of TTEO. Morphology and energy-dispersive (EDX) analysis after the immersion in simulated body fluid (SBF) exhibited a light layer of calcium chloride and sodium chloride generated on the material’s surface, which is generally related to a bioactive material. Finally, the biocompatibility of the films was comparable with porcine collagen, showing better signs of resorption as the amount of TTEO was increased. These results indicate the potential application of the films in long-term biomedical needs.

## 1. Introduction

Several studies in the literature report applied films and membranes to treat and recover damaged tissues, support cell adhesion and growth factor liberation, and provide structural support to tissues and organs [1,2]. It is of great importance that the developed films are biocompatible, porous, and biodegradable to resemble the extracellular matrix (ECM), efficiently influencing the surrounding cell population’s physical, chemical, and biological environment [3].

Biopolymers have been clinically used for several years due to their stimulating interactions with different cells and lack of immune response [4]. However, the main drawbacks of biopolymers are their low mechanical resistance and poor barrier to water, which challenges the synthesis of new materials. On the other hand, synthetic polymers are cheaper. Previous studies have focused on the study of nanocomposite polymers to overcome the disadvantages of natural polymers. Nevertheless, synthetic polymers, in general, do not stimulate cell adhesion and proliferation. However, the combination of synthetic with natural polymers represents a convenient way to solve adversities through functionalization with specific ligands, proteins, and other molecules, improving cellular responses [5]. One of the most used synthetic polymers in biomedical applications is polyvinyl alcohol (PVA), based on the ability to interact with water, non-toxic properties, and biodegradability. Furthermore, the flexibility of the PVA chain significantly contributes to its biomedical versatility, despite the instability in an aqueous environment [6]. 

Chitosan (CS) is a biopolymer obtained from the chemical modification of chitin, the main component of crustacean shells. It is a linear binary heteropolysaccharide composed of glucosamine linked to β-1,4-N-acetylglucosamine widely used in clinical applications thanks to its harmless and even stimulating interaction with tissues and organs [7]. CS has broad applications within tissue engineering, among which bone and subdermal tissue engineering with good adhesion and cell proliferation stand out. CS has been evaluated with different materials such as nanofillers, ceramics, and polymerics with excellent biocompatibility and cell proliferation results. However, the low solubility and low mechanical resistance in body fluids are its main drawbacks to expanding medicine use.

The generation of scaffolds has been carried out through different techniques such as membranes, sponges, hydrogels, and nanofibers, searching for convenient interconnectivity and porosity, facilitating the removal of waste, and the introduction of nutrients [8]. Recently, the use of polymer-based membranes with cellular biocompatibility has been evaluated in biomodels to observe cell proliferation-related to the polymeric structure [9]. Additionally, the use of chitosan-essential oil films for antimicrobial evaluation demonstrated a determining influence of the mixture of essential oils with chitosan, which could be used for wound-healing applications [10]. 

Essential oils are volatile oily liquids characterized by lipophilic biomolecules and aromatic compounds that add aromatic properties. Usually, they present complex mixtures of compounds such as monoterpenes, polyphenols, terpenes, sesquiterpenes, ketones, and aldehydes. Some essential oils components present excellent biological properties such as eugenol, cinnamaldehyde, carvacrol, and thymol [10,11], which are used for fungal, virucidal, insecticidal, antiparasitic, and antibacterial activities [12]. Previous studies have shown that cinnamon extracts [13], bergamot [14], and green tea [15] improved the mechanical properties as well as the hydrophobicity and water vapor barrier of chitosan films. However, other oils such as oregano [16], tea [17], thyme, and cloves [18] had adverse effects on the mechanical properties. *Melaleuca alternifolia* or tea tree essential oil (TTEO) is a complex mixture of different volatile compounds such as terpinene-4-ol (≥30%), γ-terpinene (about 20%), α-terpinene (about 8%), *p*-cymene (about 8%), α-pinene (about 3%), terpinolene (about 3%), and 1,8-cineol (≤15%) that have shown a potent biological activity [19]. TTEO also presents antioxidant activity, making it a good candidate for cosmetics and the food industry despite the low number of reports [20]. On the other hand, the antibacterial activity of TTEO against bacteria and yeasts in ground meat against *listeria monocytogenes* has been reported at a concentration of 1.5% (*v*/*w*) [21]. 

Likewise, TTEO has successfully been used in the clinical field to treat oral candidiasis in patients with AIDS [22] and other oral fungal infections in patients with advanced cancer [23], taking advantage of its potential pharmacological applications. Despite the reported applications of chitosan films with TTEO, no studies have been found evaluating biocompatibility under in vivo conditions for CS/PVA/essential oil compounds. For this reason, this work will evaluate the potential for tissue engineering applications of CS/PVA/TTEO to combine and harness their biological activities and improve the biocompatibility and mechanical properties of CS/PVA films.

## 2. Materials and Methods

### 2.1. Materials

Chitosan (CS) from shrimp shells with low molecular weight (*Mv* 144,000) was purchased from Sigma-Aldrich (Sigma-Aldrich, Palo alto, CA, USA) with a deacetylation degree of 90%. *Mv* was measured by capillary viscometry using a Ubbelohde 0C viscometer in 0.3 M acetic acid + 0.2 M sodium acetate at 25 °C. K and a constant (0.074 mL/g and 0.76) were used to calculate the *Mv* through the Mark–Houwink–Sakurada equation (Equation (1)) [24]. The deacetylation degree was determined by ^1^H-NMR (BRUKER AVANCE II spectrometer, 400 MHz, 300 K, Bruker, Berlin, Germany) and elemental analysis (Thermo Electron Flash EA 1112, Thermo Fischer, Waltham, MA, USA).
(1)$$\left[\eta \right]=K{\left(Mv\right)}^{a}$$

Polyvinyl alcohol (89% of hydrolysis, 93 000 g/mol) was provided by Sigma-Aldrich (Palo Alto, CA, USA). Glacial acetic acid and Tween 80 were provided by Merck (Burlington, MA, USA).

### 2.2. Composition of Tea Tree Essential Oil (TTEO)

Tea tree essential oil (*Melaleuca alternifolia*, TTEO) was purchased from Marnys (Madrid, Spain). TTEO composition was characterized by gas chromatography mass-coupled (GC-MS). C_6_-C_25_ hydrocarbons were used as the reference using an AT 6890 *series plus* gas chromatograph (Agilent Technologies, Palo Alto, CA, USA) coupled to a mass selective detector (Agilent Technologies, MSD 5975), Column DB-5MS (J & W Scientific, Folsom, CA, USA), 5%-Ph-PDMS. Identification compared RI (retention indexes) with the Adams database (Wiley, 138 and NIST05, Agilent, Santa Clara, CA, USA).

### 2.3. Preparation and Characterization of CS/TTEO Emulsions

Emulsions were prepared according to the reported methodology [25] with some modifications: two liters of emulsions were prepared by mixing 10 g of CS with a 1% (*v*/*v*) solution of acetic acid, 50 g of glycerol, and 1% of Tween 80 (concerning the volume of TTEO). After mixing for one hour with constant stirring at 30 rpm, ultrasound was applied (BRANSON, Madrid, Spain) for an extra hour at 25 °C. Finally, concentrations of 0.5%, 1.0%, and 1.5% *v*/*v* of TTEO were added concerning the amount of CS present in each emulsion. Then, a homogenizer (Manton-Gaulin MFG Co., Inc., Everett, MA, USA) was used at 3000 psi for 30 min, and at the end, the particle size was determined. Finally, the PVA was added to the previous emulsions, and they were thoroughly mixed until homogeneity and determination of their particle size.

#### 2.3.1. Particle Size

The particle size emulsions were analyzed through a previously reported methodology [26] in an AIMSIZER 2011 diffractometer following the International Organization for standardization [27].

#### 2.3.2. Viscosity Measurements

We measured the emulsion viscosities with a Brookfield LVF viscometer. Each sample was poured at room temperature into a beaker. Samples subjected to constant stirring at low speed were measured according to the ASTM D2196-99 standard [28]. 

#### 2.3.3. Total Solids

Solids were determined from the films using an oven at 150 °C until dryness using Equation (2):(2)$$\%S=\left(\frac{P_{s}-P_{d}}{P_{m}-P_{d}}\right)\times 100\;\;\;$$
where %*S*: non-volatile solids (wt %), *P_d_*: dry aluminum disk weight (g), *P_m_*: sample plus the aluminum disk weight (g), and *P_s_*: dry sample plus the aluminum disk weight (g) [26].

### 2.4. CS/PVA/TTEO Films Preparation

CS/PVA/TTEO films were prepared dissolving the PVA in the CS/TTEO emulsion up to 2% (wt %). All the components were mixed and degassed using an ultrasound bath (Branson, Madrid, Spain). The films were prepared according to previously reported methodologies [9]. The homogeneous solution was added to acetate molds and allowed to stand for 24 h in the open air. Subsequently, they were placed in a drying oven at a temperature of 40 ± 0.2 °C to obtain the solid films CS/PVA/TTEO. Samples were stored at 10% relative humidity (RH) until needed. 

#### 2.4.1. Characterization of CS/PVA and CS/PVA/TTEO Films

##### Fourier Transform Infrared Spectroscopy (FT-IR)

We identified film’s functional groups using an FT-IR in ATR (attenuated total reflectance) mode with a diamond tip (Shimadzu, Kyoto, Japan) between 500 and 4000 cm^−1^.

##### Scanning Electron Microscopy (SEM)

The film morphologies determinations used a scanning electron microscope (SEM) (JEOL JSM-6490LA, Musashino, Tokyo, Japan) with a secondary electron mode. The voltage of 20 kV was used with a gold layer for better conductivity of the samples.

##### Analysis of Mechanical Properties

Once collected, the samples were stress-tested following ASTM D6287, ASTM D618, and ASTM D882. The thicknesses were determined on a Mitutoyo No. 293–330 digital micrometer (Kawasaki, Japan) with five repetitions for each sample. The thicknesses are reported as mean ± SD (at least five measurements). Films’ tensions and elongations through (six repetitions at least) a universal test equipment SHIMADZU EZ-LZ (Shimadzu, Kyoto, Japan) under the ASTM D882 standard. Space between the jaws: 10 mm. Width film: 20 mm. Speed analysis: 50 mm/min.

##### Thermal Stability of Films

The weight loss of the films was measured through thermogravimetric analysis (TGA) on a TA instrument TGA Q50 V20.13 Build 39 instrument (TA instrument, Delaware, New Castle, DE, USA). The samples were heated to 1000 °C at a 20 °C/min heating rate under an air atmosphere (60 mL/min). On the other hand, the glass transition (Tg) and endothermic peaks were determined using differential scanning calorimetry with a DSC2A-00181 (TA instrument, Delaware, New Castle, DE, USA). Finally, the data were analyzed using the Universal Analysis software from the TA instrument.

##### Degradation in Simulated Biological Fluid

The film hydrolytic degradation of samples was performed in a simulated biological fluid (SBF) [29] following the ASTM F1635 standard. The samples were stored at 37 °C in an incubator (Memmert, Schwabach, Germany) for 1 and 16 days. The pH of solutions was recorded using an Accumet™ AB150 pH meter (Fisherbrand, OTT, Canada). Sample weights before (*W*_0_) and after immersion in SBF and dried (72 h at 37 °C, *W_d_*) were determined. Therefore, the weight loss (*W_l_*) and the water absorption (*W_a_*) of samples were determined by Equations (3) and (4), respectively. Scanning electron microscope (SEM) images were obtained after 1 and 16 days of sample immersions in SBF using a scanning electron microscope (SEM) JEOL Model JSM 6490 LV coupled to an X-ray Energy-Dispersion Spectrometer (EDS) (Akishima, Tokyo, Japan).
(3)$$W_{l}\;\left(\%\right)=\frac{W_{0}-W_{d}}{W_{0}}\times 100$$
(4)$$W_{a}\;\left(\%\right)=\frac{W_{w}-W_{d}}{W_{d}}\times 100$$

##### In Vivo Biomodel Evaluations

Following the UNE 10993-6 standard (Biological evaluation of medical devices-Part 6: Tests related to local effects after implantation, ISO 10993-6: 1994), subdermal implantations were carried out in the dorsal area of five Wistar rats (Rattus norvegicus domestic) four months old with an average weight of 380 g.

The biomodels were sedated by an intramuscular ketamine 30 mg/kg and Xilacina 70 mg/kg (Xilaxyn-Virbac., Bogotá, Colombia). The level of sedation was verified to perform the trichotomy test on the entire dorsal surface. The dorsal surface was disinfected with iodine solution. Then, lidocaine 2% with epinephrine 1:80,000 (Newcaina, Guarne, Antioquia, Colombia) was applied as a local anesthetic in the implantation zones. After that, using a bard parker scalpel with blade No. 15, six incisions of 10 cm long were made, and six pockets of 3 cm deep were created using a dental periosteal. Finally, five experimental CS/PVA/TTEO films and collagen films (control) were implanted (10 × 5 × 1 mm) films.

After 90 days of implantation, euthanasia procedures were performed by 150 mg/kg of sodium pentobarbital (Euthanex-INVET, Medellín, Colombia). Samples were recovered and fixed in buffered formalin by 48 h.

Subsequently, the samples were fragmented to preserve only the areas of interest. After that, they were dehydrated by immersion in ascending alcohol (70%, 80%, 95%, and 100%). Finally, samples were diaphanized with xylol and infiltrated with paraffin using the Autotechnicon Tissue Processor™ equipment (Leica Microsystems, Mannheim, Germany). The paraffin blocks obtained were used for histological processing with the Thermo Scientific™ Histoplast Paraffin™ equipment (Fisher Scientific, Waltham, MA, USA). Then, 5 µm sections of the films were made using a Leica microtome ((Leica Microsystems, Mannheim, Germany) and deposited on glass slides. After 48 h, they were stained with hematoxylin–eosin and Masson’s trichrome to finally take photomicrographs with a Leica optical microscope equipped with an imaging suite (Leica Microsystems, Mannheim, Germany).

The biomodels were supplied by the LABBIO laboratory of the Universidad del Valle in Cali, Colombia, and they remained in this place throughout the implantation period. The Ethics Committee carried out the authorization and ethical supervision of the Project with animals in biomedical experimentation—CEAS of the Universidad del Valle through resolution CEAS 012-019.

##### Statistical Analysis

The emulsion characterization, mechanical properties, and hydrolytic degradation were presented as a mean value of at least three replicate ± SD. We used an analysis of variance (ANOVA) and Tukey method with a confidence level of 95% (α = 0.05) to evaluate the significance in changing formulations on the responses of emulsion physical–chemical characterization, mechanical properties, and hydrolytic degradation. Statistical analyzes used the Statgraphics Centurion XVI program.

## 3. Results

### 3.1. Tea Tree Essential Oil Characterization

GC-MS analysis of TTEO showed the presence of 52 compounds (Appendix A). The compounds include 1 monoterpene, 10 hydrocarbons, 8 monoterpenes, 27 oxygenated sesquiterpenes, 5 compound mixtures, and 1 unidentified compound. Usually, monoterpenes and sesquiterpenes represent the main compounds in essential oils. However, in TTEO, oxygenated terpenes and hydrocarbon monoterpenes represent the majority and are distributed as follows: terpinene-4-ol (≥30%), γ-terpinene (about 20%), α-terpinene (about 8%), *p*-cymene (about 8%), α-pinene (about 3%), terpinolene (about 3%), and 1,8-cineol (≤15%) [19,30]. Our results are consistent with previous characterizations [30].

The TTEO is a member of the Myrtaceous family originally from New Wales and Australia. Australian ancestors used the tea tree as an antiseptic. Clinical studies showed antifungal, antiseptic, germicidal, and antibacterial properties. The most abundant compound, terpinene-4-ol, is responsible for relieving the signs of acne, psoriasis, superficial burns, and healing superficial wounds by cuts while protecting from infections. 

### 3.2. Physical–Chemical Characterization of the Film-Forming Emulsions (FFE)

To prepare homogeneous and resistant films, the film-forming emulsions must be stable for a reasonable time. Different parameters, such as the preparation, characterization, and stability, are crucial to determining emulsions’ final behavior, including bioavailability, dissolution, stability, and content uniformity in colloidal systems. Table 1 shows the characterization of the non-volatile fraction constituted by CS/PVA/TTEO. The particle size analyzer is an efficient and non-destructive method used to characterize particles and evaluate nano-sized particles’ size distribution and minor suspensions in a liquid [31].

The emulsions remained without separation after preparation, which was probably due to a small particle size, indicating that the phases do not separate within the emulsion. Generally, the emulsions analyzed in this work contain a mixture of water–essential oil–chitosan and an emulsifier (Tween 80). Emulsifying agents are responsible for stabilizing emulsions, allowing the interaction of immiscible systems through their lipid and lipophilic structure. These materials, which are wetted with water and oil, must be smaller than the emulsion droplets and accumulate at the water/oil interface. The effectiveness of these particles in stabilizing emulsions is highly dependent on several factors, such as interactions between particles, the size of the droplets, and the wettability of the material. In this regard, the emulsions prepared during the investigation did not vary considerably in their particle size, which facilitates that the phases do not separate. Additionally, after two years of preparation, the particle size of the emulsions did not vary significantly, as shown in Appendix A [32,33], indicating that our emulsification procedure was adequate to form very stable emulsions crucial for high-quality film preparation. 

No significant differences were found in the emulsion particle sizes. As can be seen in Table 2, the D_50_ of the particles was below 1.6 µm. The incorporation of TTEO did not significantly affect the particle size of the emulsion drops, contributing to the stability of the emulsions. This particle size can benefit homogeneous and non-cracked film formation [9]. In addition, small particles produce less sedimentation, and therefore, they can remain more in suspended form. This phenomenon is aided by chitosan, which stabilizes the emulsion as a colloidal protector by adsorbing oil drops electrostatically at the interface, preventing the flocculation of the oily phase and the formation of cream. 

The decreasing viscosity with the essential oil increase reduces the oily phase’s agglomeration thanks to the emulsion’s stability [34]. Probably two opposite effects contribute to the observed trend: increasing the dispersed phase (oil) increases the apparent system viscosity, while polymer adsorbed on the drop’s surface decreases the availability of CS in the aqueous phase.

### 3.3. FT-IR from CS/PVA/TTEO Films

Figure 1 shows the FT-IR characterization of CS/PVA/TTEO films. For F1, the characteristic asymmetric tension bands of the -OH and CH_2_ groups around 3302 and 2941 cm^−1^ typical of PVA are observed [11]. The tension bands overlap the stress vibration of the -OH group and the symmetric and asymmetric stress vibration of the CS NH_2_ groups [35]. In addition, CS tension bands in 1728, 1649, and 1255 cm^−1^ are appreciated. Those bands correspond to the -C=O group vibration for an amide I and III, the NH group and NH_2_ bending band, and the undulation of the CH_2_ groups, respectively. Likewise, the oscillation band at 1429 cm^−1^ is observed for the OH and CH groups of chitosan [36].

On the other hand, films incorporating TTEO (F2-F4) exhibit shifting at high wavenumbers of various bands, especially the tension band at 3302 cm^−1^ of the -OH group. The plausible cause is hydrogen bonding of -OH groups of TTEO compounds (mainly terpinene-4-ol) and chitosan [37]. Films containing turmeric essential oils have previously been observed [38], and they present a similar pattern of band shifting. In addition, an increase in the CH group band at 2941 cm^−1^ was observed for F3 and F4, suggesting an increase in ester content [17].

With the introduction of more TTEO, the band of C-C vibrational stretching (2362 cm^−1^) was evident from phenyl ethynyl groups of TTEO [39]. The band between 1464 and 1385 cm^−1^ corresponds to the bending vibration of the CH group. The increase in the band’s intensity of 856 cm^−1^ as the TTEO concentration increases is due to the symmetric spatial arrangement in the polysaccharides [10], indicating a possible change in the three-dimensional polymer chain arrangements. Finally, the asymmetric stretching of C=O and OH bands can be observed at 1112–1114, 1054–1056, and 856 cm^−1^ in all formulations, which is characteristic of the glycosidic bond chitosan. 

### 3.4. The Tensile Strength of the CS/PVA/TTEO Films

A material’s biomedical applications demand good mechanical properties (tensile strength and elongation at break) and biocompatibility. Table 2 shows the mechanical behavior of the different CS/PVA/TTEO films evaluated by the Young modulus and their tensile strength, which measures the toughness and stiffness of the studied material or its elastic capacity. The higher the modulus, the stiffer the material. Conversely, materials with low values are easier to bend under an applied force. Finally, it is essential to remember that the modulus of elasticity is the relationship between the material’s stress and unit deformation and represents its rigidity before imposing a load [40].

The influence of TTEO on the mechanical properties of the film differs from previous works [41,42]. In this regard, the intramolecular interaction between the components affects the mechanical and thermal properties, reflecting on the type of bond (hydrogen bonding, electrostatic interaction, van der Waals forces, among others) and the interactions with the components TTEO (dispersion). In turn, this dispersion is affected by the relative humidity, surfactants, and system temperature, leading to significant particle dispersion changes in the films [17,35].

Additionally, the introduction of TTEO in the films significantly (*p* < 0.05) affects the tensile strength in the sheets. Table 2 shows an increase in the tension force from 5.5 ± 1.0 to 6.3 ± 0.7 Pa between F1 and F2. Furthermore, a decrease in the percentage of deformation between 362.0 ± 25.5 and 280.9 ± 12.9 Pa was observed for these formulations. However, as the TTEO incorporation increases, the tensile strength and deformation of the films decrease, with F4 being the one with a significant decrease in the mechanical properties of resistance to tension and deformation of the film (4.6 ± 0.6 and 188.2 ± 13.8, respectively). 

The relationship between the tensile strength and elasticity of the films was evaluated through Young’s modulus. An increase of 1.6 ± 0.3 to 2.0 ± 0.2 Pa is observed between the F1 and F2 films. Probably, the new hydrogen bonds between the -OH groups of the polymers and the TTEO are the plausible cause, as shown in the FT-IR [17]. Meanwhile, the addition of 1.0% and 1.5% TTEO significantly decreases (*p* < 0.05) Young’s modulus to 1.40 ± 0.14 and 1.58 ± 0.08 Pa for F3 and F4, respectively. Probably, the discontinuities generated by the flocculation of TTEO in the film cause lower resistance to fracture [12]. This observation is consistent with other authors when adding essential oils to the CS matrix [16,41]. 

### 3.5. Thermal Analysis of CS/PVA/TTEO Films

The effect of TTEO incorporation on the thermal resistance of the CS/PVA films was evaluated by TGA and DSC, as shown in Figure 2 and Figure 3. CS and PVA decompositions are shown in the Appendix A. For PVA (Appendix A), three stages are evident. The first weight loss from 50 to 170 °C is also related to solvent evaporation. The second weight-loss peak at 230 °C is originated from the lateral groups of PVA. Polyene residues, after degradation, originate the third stage of weight loss at 400 °C.

The first multiple signals in a wide range of frequencies in CS decomposition beginning between 50 and 110 °C (Appendix A) are related to solvent evaporation. The second stage begins at 220 °C, which is caused by the dehydration of the material. Finally, the third stage, which begins at 320 °C, is related to losing glycosidic bonds in chitosan [37]. 

Regarding the films, previous studies reported the thermal degradation corresponding to chitosan films with essential oil in three stages [43]. However, four stages of thermal degradation were observed with the incorporation of TTEO in formulations F2, F3, and F4 (Figure 2B–D). Similar behavior was observed concerning the degradation ranges between F1, F2, and F3 (Figure 2A–C). The first weight loss is about 50–100 °C due to water and acetic acid traces evaporation. Fractions of CS degraded at 190–250 °C, and there was also a loss of volatile essential oil compounds and decomposition of the PVA side chain. The third degradation stage at 250–300 °C reflects the degradation of the saccharide rings and decomposition of the acetylated and deacetylated units of the CS. Finally, the fourth stage begins between 440 and 500 °C and is associated with the polyene residues from the PVA.

On the other hand, the degradation between 50 and 100 °C of the films was not observed to occur for F4 (Figure 2D). The disappearance of the weight loss might be a result of a more hydrophobic character of the film. Additionally, between 300 and 350 °C, a new weight loss peak was raised, which was probably associated with volatile components of TTEO such as monoterpenes and sesquiterpenes from the polymer matrix [14]. The values of Td_3%_ (3% weight loss) for all the films are shown in Table 3. For F2, F3, and F4, an increase was observed for Td_3%_. A higher dispersion in the polymer matrix of the oil, which generates hydrogen bonds between the CS/PVA and TTEO chains, and a more hydrophobic structure (less free water) could be the plausible cause. 

DSC measurements (Figure 3) can provide information on the material’s thermal behavior and phase transitions. Two endothermic peaks can be observed for F1 at 155 and 191 °C, where the endothermic peak at 155 °C can be related to the residual evaporation of the solvent. With the incorporation of 1.0 and 1.5% TTEO (F3 and F4), no significant differences in the temperature are observed in endothermic peaks. However, a narrower and more intense peak might indicate changes in the crystallinity of the structure because of the higher TTEO amount. On the other hand, the peak displacement at higher temperatures when 0.5% TTEO is added to the F2 formulation suggests a significant intramolecular interaction in the film between the better dispersion of the oil components. Chemisorbed water bonds and amino groups of chitosan interact through hydrogen bonds [16,44], with the oil components improving the dispersion through the polymer matrix. That improved dispersion increases the film’s mechanical and thermal properties due to the decreased porosity and increased compaction of the polymeric chains. Similar behavior was observed when α-tocopheryl acetate [45], citronella essential oil, and cedarwood oils [46] were added to chitosan films. 

### 3.6. Scanning Electron Microscopy (SEM) of CS/PVA/TTEO Films

Surface morphology studies of CS/PVA/TTEO films can be seen in Figure 4. The scanning electron microscopy (SEM) technique supports the analysis of the films’ microstructure and is helpful to understand the relationships between structural characteristics and physical properties (mechanical, thermal, and optical properties). F2, F3, and F4 showed a smooth surface without cracks or breaks due to the presence of TTEO in the entire matrix, indicating that the homogenization of the films allowed the elimination of excess bubbles produced by agitation and the complete dispersion of the oil in the polymer matrix.

On the other hand, irregularities in formulations F3 and F4 (Figure 4E–H) increased with TTEO incorporation due to the reduced mobility of the TTEO drops induced by the highly viscous chitosan solution [47]. In addition, the 1.5% concentration of TTEO showed pearl formation, possibly at the high oil concentration, which facilitates the aggregation of lipids, leading to a flocculation of TTEO in the polymer that occurred during the volatilization of the solvent in the drying process. 

### 3.7. CS/PVA/TTEO Film Degradation in Simulated Body Fluid (SBF)

#### 3.7.1. Weight Loss

Figure 5 shows the weight loss (%) of the composite films after 16 days in SBF. In general, TTEO increased the film stability, as reflected by a lower weight loss after seven days of immersion (F1 72%, while F4, 70). Additionally, a more significant difference was found at 16 days, where 82% weight loss was elucidated for F1, while 77% was observed for F4. This behavior is because of the excellent dispersion of TTEO in the CS/PVA polymer facilitated by several non-covalent interactions such as hydrogen bonding and van der Waals interactions, providing high chemical stability to degradation and higher hydrophobic nature, reducing water incorporation.

#### 3.7.2. pH Changes

Figure 6 shows the pH changes in SBF during the sample immersion. The degradation of amorphous zones of CS releases acidic groups such as glucosamine and N-acetylglucosamine, contributing to the acidic pH [5]. Examination of the human extracellular matrix demonstrated those CS by-products exhibiting good biocompatibility [48]. On the other hand, for formulations F3 and F4, a turning point is found approximately at day 5, increasing the pH, which is probably due to the consumption of organic acids due to the interaction with the SBF, proliferating calcium and sodium ions. These observations are consistent with the SEM after the immersion experiments. Maintaining the physiological pH in tissues between 6 and 8 requires enzymes to be functional in biological processes, which was accomplished in the present study with our samples [49]. 

#### 3.7.3. SEM of Films after Immersion in SBF

SEM images of the CS/PVA/TTEO films after immersion in SBF are shown in Figure 7. Film morphologies show notable differences before and after immersion due to the salts’ degradation and deposition process, such as the presence of apatite. In fact, the change in the film morphologies with globular deposition characteristics with nucleation sites are traits in bioactive materials with the presence of apatite [46,50]. The apatite layer adsorbed on the film’s surfaces makes the films more bioactive and promotes cell adhesion. On the other hand, small layers belonging to calcium chloride and sodium chloride were observed by energy-dispersive X-ray spectroscopy (EDS) with a calcium content of 22.2% and sodium content of 2.61% for formulation F4. Additionally, formulation F2 containing 0.5% of TTEO, which increases mechanical (2.0 ± 0.2 Pa) and thermal (Td_3%_ 61 °C) properties, showed a sodium content of 2.56%. Therefore, it might be reasonable to hypothesize that a higher content of TTEO promotes the affinity of the films for apatite layers. 

## 4. In Vivo Biocompatibility Tests of the CS/PVA/TTEO Films

Biocompatibility and resorption studies of the samples after 90 days of implantation followed the UNE 1093-6 (ISO 10993-6) standard. The standard suggests that once euthanasia has been carried out, a macroscopic inspection of the biomodels should be carried out in the intervened areas. During this inspection, the skin on the dorsal surface was healthy and with hair recovery (Figure 8A).

The biomodels underwent a trichotomy to remove hair, and tissue recovery was observed in the intervened areas (Figure 8B). On the other hand, a midline incision was made, and the flaps were separated to review the internal area of the skin, where areas with slight color changes corresponding to the implantation areas and the touch were observed. Slight bulges corresponding to the implantation areas were also observed. Figure 8C shows the area corresponding to preparation 1 with a more intense color. Finally, it was observed that all areas were covered by healthy tissue and without necrosis, which suggests the high biocompatibility of the material with the implanted tissue (Figure 8C).

Once euthanasia had been performed, the samples were recovered, and histological studies were carried out for staining with hematoxylin–eosin and Masson’s trichrome techniques.

The problems in processing samples were reflected in the histological images where folding and displacement of the samples were observed. F1 and F2 films presented complex processing as a tear during the cut during the histological sections due to the high mechanical strength of the films. However, samples F3 and F4 were easier to process. Numerous fragments of F1 are observed in the implantation zone in the middle of white spaces that correspond to the displacement of the samples during cutting (for example, in Figure 9B).

A fragment was selected (yellow circle in Figure 9A and black circle in Figure 9B) to analyze the resorption process of sample F1. Figure 9B shows that the sample underwent a fragmentation process, and each fragment is in the middle of an inflammatory infiltrate. Here, the fragment is perfectly delimited (Figure 9B). At higher magnification (40×, Figure 9C), we can appreciate the fragment surrounded by a fibrous capsule and in the process of resorption. It is also observed that the fragment shows irregular lateral areas due to the beginning of the resorption process.

With sample F2, a similar situation was observed with the presence of numerous non-reabsorbed fragments due to the high stability provided by polyvinyl alcohol in the films (Figure 10A). The fragments are embedded in an inflammatory infiltrate, as seen in Figure 10B. Figure 10C (40×) shows the fragment marked with an oval in Figure 10A in the middle of an inflammatory infiltrate but delimited by what appears to be a fibrous capsule. With the previous evidence, we can conclude that samples were biocompatible during the 90 days of implantation, but with slow reabsorption, thanks to the high mechanical stability and body fluids that the films presented because of the incorporation of polyvinyl alcohol in the formulations.

Sample F3 showed more significant evidence of reabsorption through irregularities on the surface of the implanted material, which is clearer at higher magnifications. Figure 11 shows some sample fragments in the middle of an inflammatory infiltrate (II, Figure 11). At higher magnification (Figure 11B,C), different size fragments are observed in the process of resorption. These fragments present irregularities in the surface morphology, which indicates that these samples were reabsorbed. The previous results indicate that the increase in TTEO stimulates the reabsorption of the material and promotes its biocompatibility in the subcutaneous tissue. 

Sample F4 was also observed fragmented after 90 days of implantation in subdermal tissue (Figure 12A), surrounded by an inflammatory infiltrate (Figure 12B) and in the process of reabsorption (Figure 12C). Both the F3 and F4 samples showed more significant signs of resorption, indicating that the incorporation of the oil promotes more significant resorption and biocompatibility. It is likely that the higher hydrophobic character of the films with a higher content of TTEO may have a better interaction or a stimulating effect for a higher production of platelets and the components of the coagulation cascade that repair skin lesions, which are fundamentally composed of fibrin, factors of coagulation, and platelets of protein character.

The porcine collagen film used as positive control showed almost complete material resorption without evidence of fragments (Figure 13). However, an abundant lymphocytic-type infiltrate is observed throughout the implantation area. 

The same sections processed with Masson’s trichrome technique are reviewed, allowing a histopathological wound-healing study. Masson’s trichrome stain can differentiate important morphological keys for the evaluation of wound healing, such as keratin presence, hemoglobin presence, muscle fiber formation (red color), cytoplasm and fat cells (light red or pink), cell nuclei (brown dark to black), and collagen fibers (blue-stained). The latter can be measured using image analysis software. The wound healing understanding from the histopathology studies is vital to understanding the recovery mechanism and the immune response [51]. Figure 14 confirms the presence of a fibrous capsule composed of collagen bundles type I around the implanted samples, where the resorption mechanism appears to be initiated by fractionation of the samples. Subsequently, each fragment is surrounded by a lymphocytic-type inflammatory infiltrate circumscribed by collagen capsule type I.

The fragment immersed in the inflammatory infiltrated and surrounded by the fibrous capsule was observed for all the samples analyzed, which shows the outstanding biocompatibility of these composites. The histological image for samples F1 and F2 is very similar. At a magnification of 10×, the two samples remain as long fragments with smaller fragments (Figure 14A,C) without significant signs of resorption on the surface or in the lateral areas. However, the 40× images (Figure 14B,D) confirm the low resorption of the material due to the low content of TTEO.

More significant reabsorption is observed with a higher incorporation of TTEO (F3, TTEO 1.0 wt %). Figure 14E shows a small fragment in the process of resorption (area marked with a yellow oval). With a higher magnification (40×, Figure 14F), fragments of cylindrical shape are observed due to the action of the phagocytic cells surrounding it in a centripetal reabsorption process already reported for other materials [9].

Figure 14G corresponds to F4 (CS/PVA/TTEO 1.5 wt %) at 10×. This figure shows fragments of different sizes in the degradation process at higher magnification (40×, Figure 14H). It is possible to see irregular borders because of degradation by phagocytic cells, which shows that incorporating TTEO stimulates a more significant interaction with these cells and a faster degradation that promotes their reabsorption in the subcutaneous tissue.

On the other hand, the collagen film used as a positive control material when processed with Masson’s trichrome technique also shows collagen fibers in the implanted area. Likewise, no material remains are observed, but the persistence of an inflammatory infiltrate of the lymphocytic type with the recovery of the tissue architecture and adipocyte cells characteristics of the hypodermis were observed in the implantation area. This result confirms that CS/PVA/TTEO films with a higher content of TTEO (F3 and F4) are reabsorbed faster than those that do not have (F1) or have a lower TTEO content (F2). This reabsorption is slower than for porcine collagen, demonstrating that these materials could be applied in the regeneration of tissues requiring a longer implantation time, as occurs in bone tissue applications.

All samples appear immersed in a lymphocytic-type inflammatory infiltrate and surrounded by a fibrous capsule composed of collagen fiber type I bundles. This typical response corresponds to a reaction to a foreign body, which our group had previously reported in subdermal implantations of chitosan and graphene oxide films [52].

The foreign body reaction is one of the mechanisms used by the organism in the presence of a biomaterial whose physical or chemical characteristics make the degradation process by phagocytic cells difficult [53]. Anderson, Rodríguez, and Chang include it within the sequence of events after implantation and describe how the characteristics of the material determine how the organism will resolve the situation of the foreign body [54]. The final processes of the foreign body reaction are also affected by the origin of the material. Once implanted, materials of biological and synthetic origin incite the same foreign body reaction process. However, those of biological origin overcome the process more quickly, leading to an early recovery of the tissue, while those of synthetic origin can stimulate the inflammatory response for a long time due to the delay in its degradation [55]. Therefore, a greater incorporation of TTEO, a natural extract, could stimulate faster degradation and reabsorption (although slow compared to porcine collagen). 

During the implantation process, tissue damage and a breakdown of homeostasis occurred. The natural healing process recovers homeostasis, going through four stages: hemostasis, inflammation, proliferation, and remodeling. In this process, there is the active participation of inflammatory response cells and reparative cells that will produce healing with average tissue recovery or impaired healing [44].

In this work, collagen films with indicated use in tissue regeneration were used as the control material. The results showed that at 90 days (Figure 15), the material is almost completely reabsorbed, but an inflammatory infiltrate of the lymphocytic type persists that may be indicating the presence of particles of the material not identifiable by the techniques used. However, being a biocompatible material, the injury caused during implantation is resolved through a healing process that recovers the tissue architecture without leaving scars or other alterations.

The four samples implanted at 90 days showed partial resorption of the materials, which incites the permanence of a slight inflammatory response, of the lymphocytic type and the presence of the fibrous capsule, but with the recovery of the tissue architecture at a macroscopic level, without significant alterations at the microscopic level as evidenced by histological studies, which would be an indicator of the biocompatibility of the implanted CS/PVA/TTEO films.

The biocompatibility shown by the implanted films could be provided by chitosan, which is a material recognized for having properties such as biocompatibility and biodegradation/reabsorption, stimulating healing, and producing a minimal inflammatory response [56]. Additionally, this biocompatibility and more significant reabsorption are stimulated by incorporating TTEO, which shows how essential oils are suitable to improve the assimilation of this type of materials and promote their application in tissue engineering [9]. 

## 5. Conclusions

The present methodology’s CS/PVA/TTEO films demonstrated high thermal and mechanical stability and surface homogeneity without fracture or discontinuities, resulting in SBF stability. In the FT-IR spectrum, compatibility between the components was found by forming hydrogen bonds and chemical interactions, which were reflected in the shift of some bands. Mechanical and thermal studies exhibited an increase in Young’s modulus and decomposition temperatures. Furthermore, all the samples presented an inflammatory process with the presence of a lymphatic infiltrate component 30 days after implantation, which decreased to a mild inflammatory process after 90 days, including the control sample of porcine collagen, which is considered a healthy absorption process. The addition of TTEO to CS/PVA increased the degradation time, both in SBF and in the implanted biomodels. After immersion of the SBF films, a light layer of calcium chloride and sodium chloride was generated on the material’s surface, which is generally related to a bioactive material. The F3 and F4 films containing 1.0 and 1.5% of TTEO showed more significant signs of reabsorption, which indicates that the incorporation of the oil promotes greater reabsorption and biocompatibility, stimulating a more significant interaction with phagocytic cells.

## Figures and Tables

**Figure 1 polymers-13-03753-f001:**
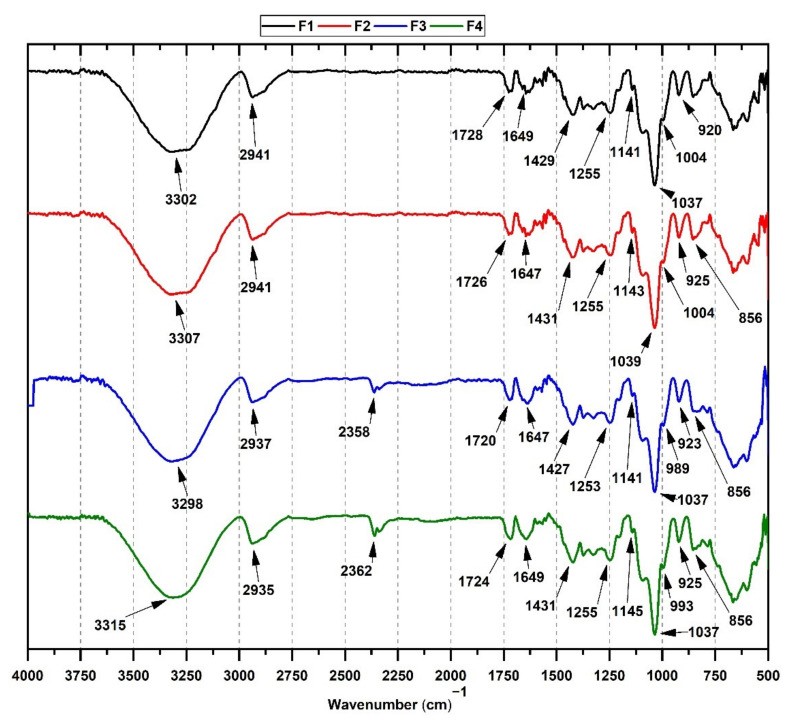
FT-IR spectrum of CS/PVA/TTEO films. F1 (CS/PVA/TTEO 30/70/0); F2 (CS/PVA/TTEO 29.5/70/0.5); F3 (CS/PVA/TTEO 29/70/1); F4 (CS/PVA/TTEO 28.5/70/1.5).

**Figure 2 polymers-13-03753-f002:**
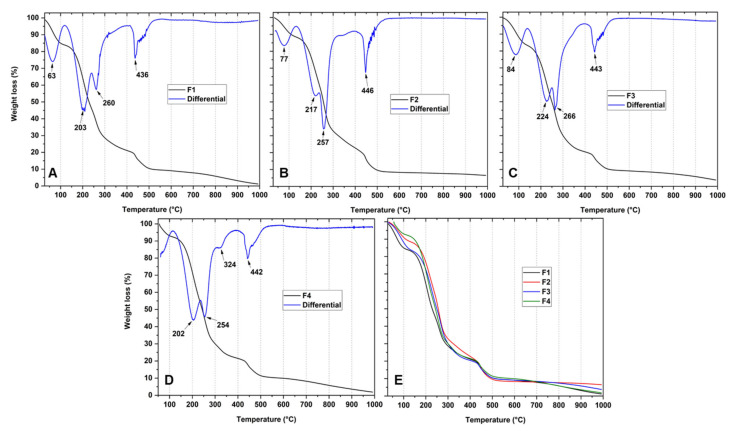
TGA curves of the films. Formulations: F1, (**A**) (CS/PVA); F2, (**B**) (CS/PVA/TTEO 0.5%); F3, (**C**) (CS/PVA/TTEO 1%); F4, (**D**) (CS/PVA/TTEO 1.5%); (**E**) all the formulations.

**Figure 3 polymers-13-03753-f003:**
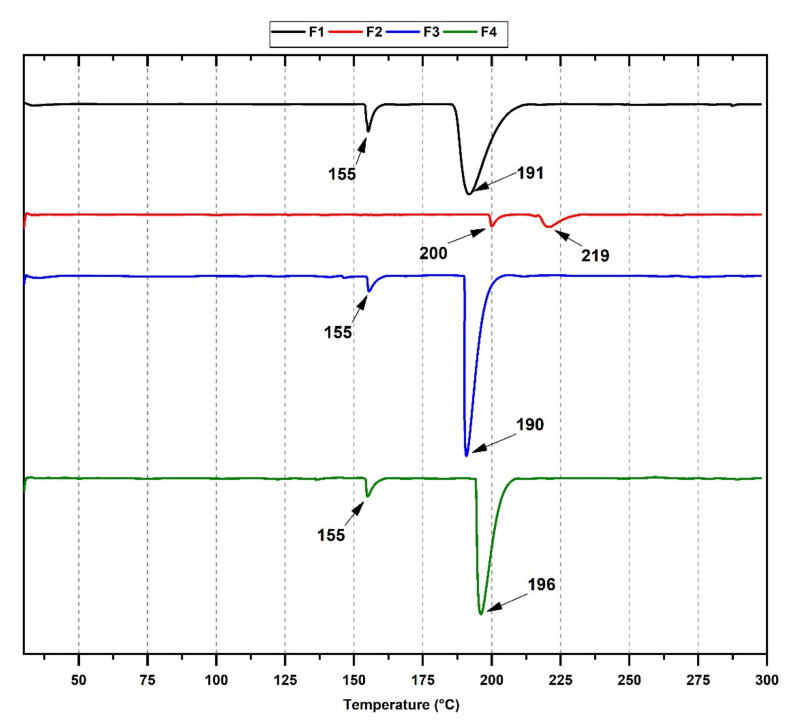
DSC curves of the films. Formulations: F1, (CS/PVA); F2, (CS/PVA/TTEO 0.5%); F3, (CS/PVA/TTEO 1%); F4, (CS/PVA/TTEO 1.5%).

**Figure 4 polymers-13-03753-f004:**
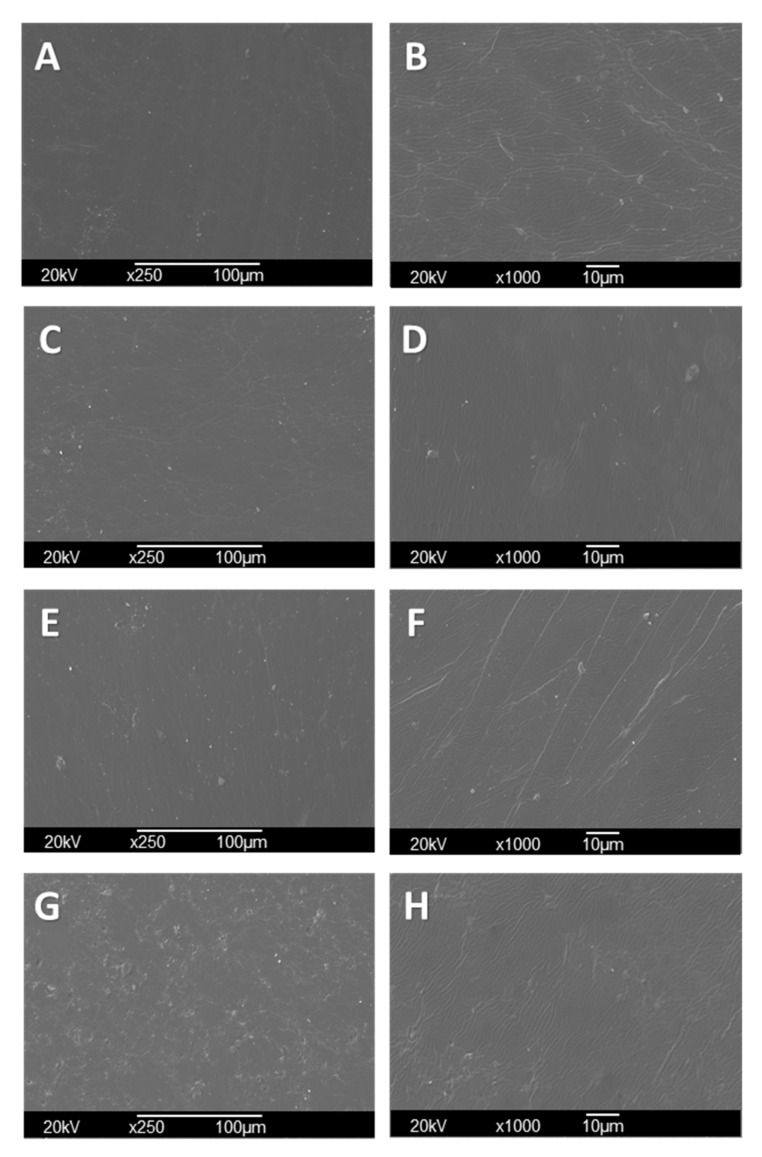
Morphology of CS/PVA/TTEO films by SEM. CS/PVA (F1) (**A**) at 250×, (**B**) at 1000×; CS/PVA/TTEO 0.5% (F2) (**C**) at 250×, (**D**) at 1000×; CS/PVA/TTEO 1.0% (F3) (**E**) at 250×, (**F**) at 1000×; CS/PVA/TTEO 1.5%(F4) (**G**) at 250×, (**H**) at 1000×.

**Figure 5 polymers-13-03753-f005:**
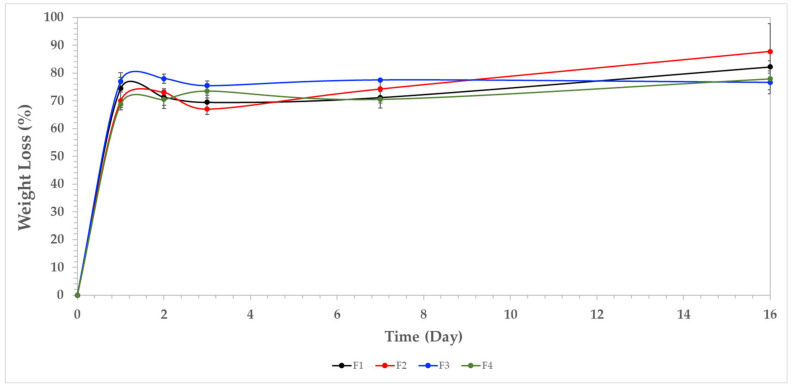
Weight loss percentage in SBF for the films. F1, (CS/PVA); F2, (CS/PVA/TTEO 0.5%); F3, (CS/PVA/TTEO 1.0%); F4, (CS/PVA/TTEO 1.5%).

**Figure 6 polymers-13-03753-f006:**
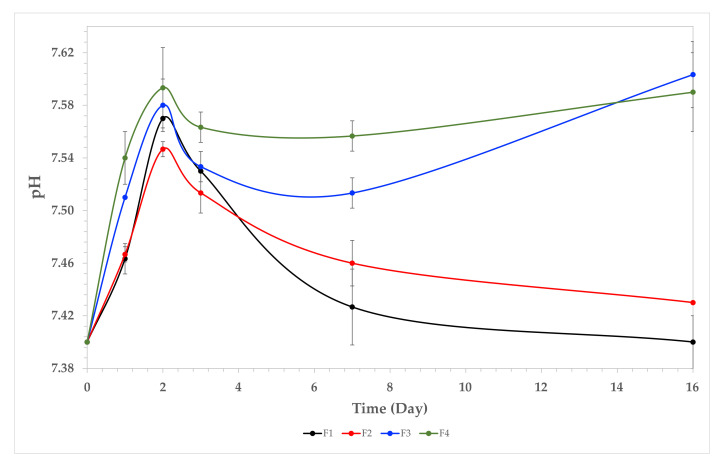
Solution pH variation vs. immersion time in simulated biological fluid. F1, (CS/PVA); F2, (CS/PVA/TTEO 0.5%); F3, (CS/PVA/TTEO 1.0%); F4, (CS/PVA/TTEO 1.5%).

**Figure 7 polymers-13-03753-f007:**
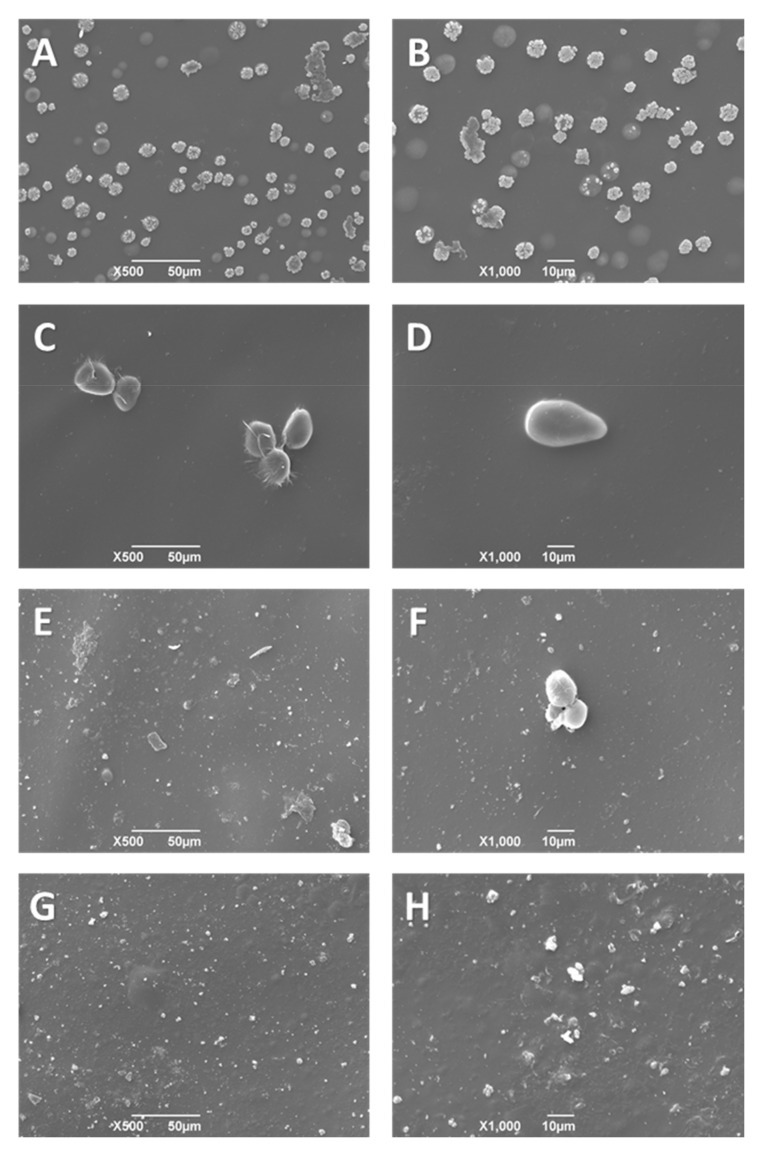
Morphology of the films after 15 days: CS/PVA (**A**) at 500×, (**B**) at 1000×; CS/PVA/TTEO 0.5% (**C**) at 500×, (**D**) at 1000×; CS/PVA/TTEO 1.0% (**E**) at 500×, (**F**) at 1000×; CS/PVA/TTEO 1.5% (**G**) at 500×, (**H**) at 1000×.

**Figure 8 polymers-13-03753-f008:**
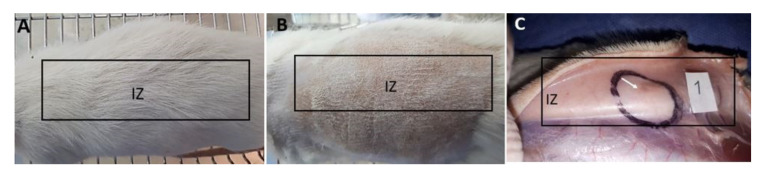
Macroscopic view of subdermal implantations. (**A**) Dorsal view. (**B**) Dorsal view with a trichotomy. (**C**) Internal area of the skin. IZ: Area of implantation. Oval: Preparation zone 1.

**Figure 9 polymers-13-03753-f009:**
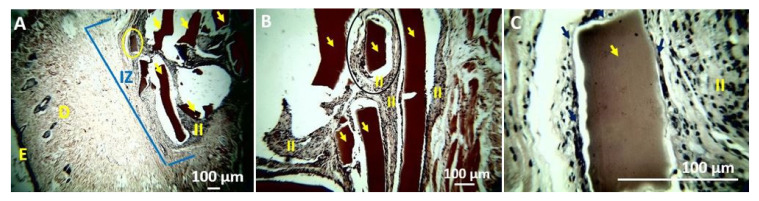
Histology of the F1 film. (**A**) 4×. (**B**) 10×. (**C**) 40×. E: Epidermis. D: Dermis. IZ: Implantation Zone. II: Inflammatory Infiltrate. Yellow arrows indicate implanted material. Blue arrows: Fibrous Capsule. Hematoxylin–Eosin Technique (H&E).

**Figure 10 polymers-13-03753-f010:**
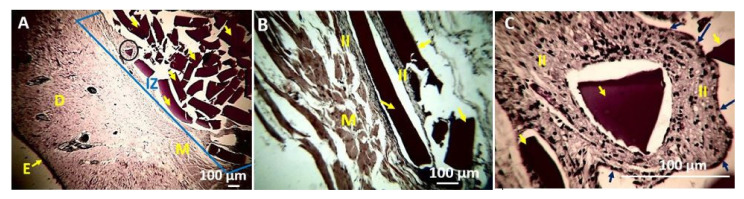
Histology of the F2 film. (**A**) 4×, (**B**) 10×, (**C**) 40×. E: Epidermis. D: Dermis. II: Inflammatory Infiltrate. M: Muscle. Yellow arrows indicate the implanted material. Blue arrows: Fibrous capsule. Hematoxylin–Eosin Technique (H&E).

**Figure 11 polymers-13-03753-f011:**
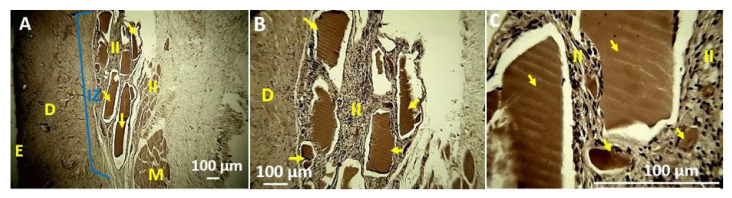
Histology of the F3 film. (**A**) 4×, (**B**) 10×, (**C**) 40×. E: Epidermis. D: Dermis. II: Inflammatory Infiltrate. M: Muscle. Yellow arrows indicate the implanted material. Hematoxylin–Eosin Technique (H&E).

**Figure 12 polymers-13-03753-f012:**
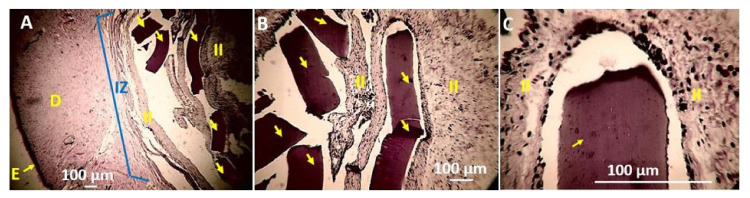
Histology of the F4 film. (**A**) 4×, (**B**) 10×, (**C**) 40×. E: Epidermis. D: Dermis. II: Inflammatory Infiltrate. Yellow arrows implanted material. Blue arrows: Fibrous capsule. Hematoxylin–Eosin Technique (H&E).

**Figure 13 polymers-13-03753-f013:**
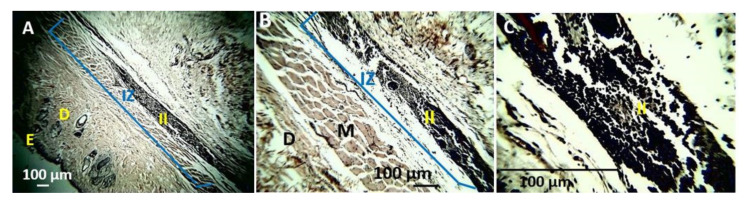
Histology of the porcine collagen film (**A**) 4×, (**B**) 10×, (**C**) 40×. E: Epidermis. D: Dermis. II: Inflammatory Infiltrate. Hematoxylin–Eosin Technique (H&E).

**Figure 14 polymers-13-03753-f014:**
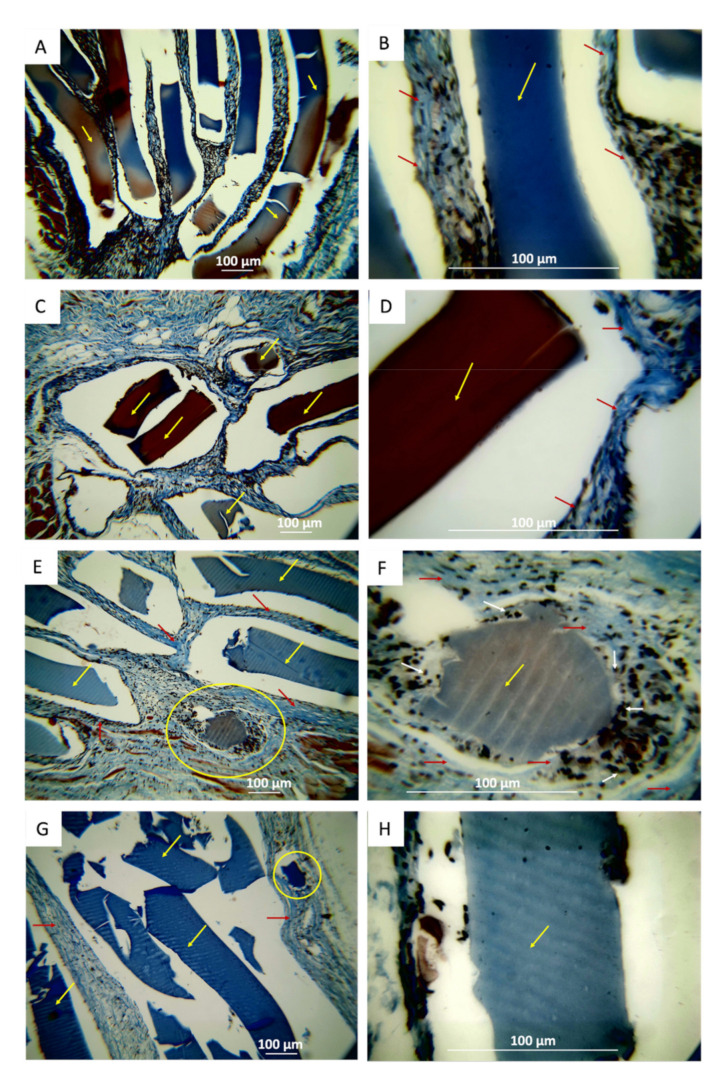
Histology of F1–F4. (**A**,**B**) F1 films. (**C**,**D**) F2 films. (**E**,**F**) F3 films. (**G**,**H**), F4 films. Images (**A**,**C**,**E**,**G**) at 10×. Images (**B**,**D**,**F**,**H**) at 40×. The yellow arrow indicates implanted material. Red arrows show the collagen capsule. White arrows indicate the presence of lymphocytes—Masson’s trichrome technique.

**Figure 15 polymers-13-03753-f015:**
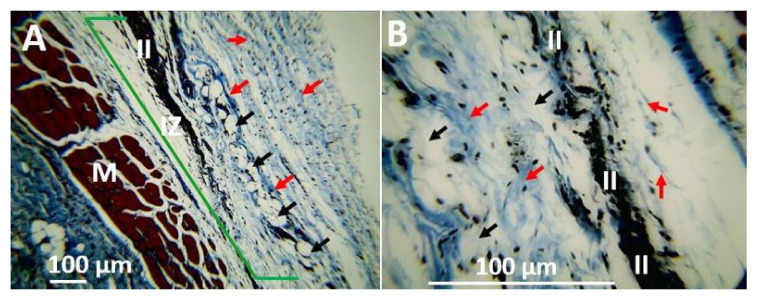
Histology of the porcine collagen film 90 days after subdermal implantation. (**A**) 10×, (**B**) 40×. IZ: Implantation Zone. II: Inflammatory Infiltrate. The red arrow indicates the presence of collagen fiber. The black arrow indicates the presence of adipocytes.

**Table 1 polymers-13-03753-t001:** Physical–chemical analysis of the CS/PVA/TTEO film-forming emulsions.

Sample	Solids (%) *	pH *	Apparent Viscosity (cP) (1/100) *	Density (g/mL)	Particle Size *	
					D_50_ (µm)	D_98_ (µm)
CS	1.66 ^b^ ± 0.03	7.43 ^a^ ± 0.009	106 ^a^ ± 0.000	1.00	N/A	N/A
CS/TTEO 0.5%	2.46 ^a^ ± 0.04	7.44 ^a^ ± 0.007	74 ^ab^ ± 0.444	1.00	1.29 ^a^ ± 0.140	3.16 ^a^ ± 0.160
CS/TTEO 1.0%	2.05 ^a^ ± 0.01	7.44 ^a^ ± 0.009	66 ^ab^ ± 0.000	1.00	1.43 ^a^ ± 0.031	3.44 ^a^ ± 0.069
CS/TTEO 1.5%	2.10 ^a^ ± 0.06	4.55 ^a^ ± 0.013	28 ^b^ ± 0.444	1.00	1.60 ^a^ ± 0.024	3.38 ^a^ ± 0.049

*** Different letters in the same column indicate significant differences (*p* < 0.05); SD (standard deviation); N/A (not applicable).

**Table 2 polymers-13-03753-t002:** Mechanical properties of CS/PVA/TTEO films.

Formulation	Young’s Modulus (Pa) *	Tensile Strength (Pa) *	Deformation (%) *
F1	1.6 ^a^ ± 0.3	5.5 ^a^ ± 1.0	362.0 ^a^ ± 25.5
F2	2.0 ^b^ ± 0.2	6.3 ^a^ ± 0.7	280.9 ^b^ ± 12.9
F3	1.4 ^a^ ± 0.1	6.0 ^a^ ± 2.3	269.7 ^b^ ± 52.3
F4	1.6 ^a^ ± 0.1	4.6 ^a^ ± 0.6	188.2 ^c^ ± 13.8

* Different letters in the same column indicate significant differences (*p* < 0.05).

**Table 3 polymers-13-03753-t003:** The 3% mass loss temperature for CS/PVA/TTEO films.

Formulations	Td_3%_ (°C)
F1	42
F2	61
F3	56
F4	68

## Data Availability

Data available under request to the corresponding author.

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
