# Peer review of "Chitosan/Polyvinyl Alcohol/Tea Tree Essential Oil Composite Films for Biomedical Applications"

_polymers, 2021, doi:10.3390/polym13213753_

Round 1

Reviewer 1 Report

This paper reports on the “Chitosan/Polyvinyl alcohol/ Tea Tree Essential Oil composite films for Biomedical Applications”. Introduction can be improve. Conclusion , methodology and reference, results and discussion seems be corrected.

I have few comments to the manuscript:

  1. All manuscript. Corrected from e.g. “[2,3]” to “[2-3]”.
  2. Reference 1 is missing in text.
  3. Tab. 1. It should be in the support materials
  4. What does 0.000 standard deviation mean?

Taking into account all comments the manuscript may be published in Polymers after minor revision.

Author Response

We are very grateful for the time spent on all the evaluator's corrections that helped significantly improve the manuscript. The suggestions are solved point by point in the attached letter and highlighted in the manuscript.

Reviewer 2 Report

The language of this manuscript is very poor. It needs to be corrected to a more scientific one. Also, there are English language problems which should be corrected before publication in any journal. For example, broadband is not used in thermal analyses but in internet connection (multiple signals at wide range of frequencies).  The novelty of this work should be stated. There are several works dealing with the topic and the references used in this work are not adequate. 

Line 60, use biopolymer instead of polyamine.

Line 70, it should be reformulated to more scientific language. Moreover, essential oils concept is quite broad. 

Line 97. Eliminate "The" . The source, purity and provider of chitosan has to be provided. Otherwise, the chemical characterization of chitosan and the isolation have to be described. 

Table 1 is very large and the composition can change due to seasonal variations. Therefore, I recommend to simplify it with the most prominent components. The rest of the components can be moved to supl. 

Line 331 what is the definition of a good stability?

Line 337, which kind of resistance. This is speculative. Moreover, the authors have not evaluated as a control a wound healing intended product. 

Line 386, change is this sense for in this regard.

Line 382 you cannot use in particular as beginning of a new phrase. 

Line 400 Young moduli cannot behave because it is a property. Please rephrase. 

The conclusions are quite simple and they do not show the knowledge obtained in this work. They have to be completed. 

The citations should be improved. They should emphasize only films made of chitosan. Example nanofibrous materials do not belong to this topic. References 8, 9, 14. The reference 15 is water purification can the authors revealed the relevance to this work? The same is true for 22 (food packaging). Reference 56 flu is not related to the topic. Please correct. There are 78 references. This is not a review then, please use the relevance work. Furthermore, relevant references were not included

Eugenol-Containing Essential Oils Loaded onto Chitosan/Polyvinyl Alcohol Blended Films and Their Ability to Eradicate Staphylococcus aureus or Pseudomonas aeruginosa from Infected Microenvironments

https://doi.org/10.3390/pharmaceutics13020195

Chitosan/Essential Oils Formulations for Potential
Use as Wound Dressing: Physical and
Antimicrobial Properties 

doi:10.3390/ma12142223

Author Response

(The authors gave the same response as above.)

Reviewer 3 Report

Tissue engineering is a challenging task of modern interdisciplinary science and technology encompassing a vast number of physical, chemical, mechanical, biological issues. In relation to this, the present work addressing the synthesis of several film composites of essential oils, and the subdermal implantations in rats is certainly on demand. There are interesting new findings showing that at 90 days the material is almost completely reabsorbed, albeit there might be particles of the material not identifiable by the techniques used. However, being a biocompatible material, the injury caused during implantation is resolved through a healing process that recovers the tissue architecture without leaving scars or other alterations. The key material providing biocompatibility is believed to be chitosan.

The strength of the work: Combination of state-of-the art sample preparation and analytical techniques providing quite convincing experimental data and their interpretation. Interdisciplinary character of the work yielding an enhanced synergistic scientific and applied effect.

The weakness: The work would gain if the data were added by molecular level information, e.g. by FTIR, Raman, etc, techniques well suited for the samples and phenomena under consideration.

In general, the work is well-motivated, scientifically sound, well conducted and arranged, clearly presented, relies on extensive background. In my view, it is suitable for publication in Polymers in its present form.

Author Response

We are very grateful for the time spent on all the evaluator's corrections that helped significantly improve the manuscript. The suggestions are solved point by point in the attached letter and highlighted in the manuscript. Also, we presented all the explanations for the characterization in terms of the film's composition (FTIR, TGA, DSC, SEM, and GC-MS for TTEO). 

Round 2

Reviewer 2 Report

Still, I do believe there are many references for an experimental paper, but it is up to the authors to correct that. The manuscript should be more focalized to tissue engineering and to chitosan chemistry.  Example reference 56 is outside this topic. The same is true for 61 (this material is intended for external application and not for bone). Reference 51 is also outside. Reference 47 is quite old, a new reference with similar chitosan (new materials) should be used. Reference 46 is also outside because, it belongs to edible films. Thus outside the topic.  Reference 39 is outside the topic (nanocellulose is not chitosan). Reference 34 is about PCL???? 

Chlorella vulgaris is also outside the topic (reference 32).

Author Response

Prof. Guest Editor Dear Editor, We want to submit our corrected version of the paper entitled "Chitosan/Polyvinyl alcohol/ Tea Tree Essential Oil composite films for Biomedical Applications" by Jorge Iván Castro, Carlos Humberto Valencia Llano, Mayra Eliana Valencia Zapata, Yilmar Joan Restrepo, José Herminsul Mina Hernandez, Diana Paola Navia, Yamid Valencia, Cesar Valencia, and Carlos David Grande-Tovar, who agreed with all the corrections. The corrections are presented below point by point in red for easy comprehension. Reviewer 2 Still, I do believe there are many references for an experimental paper, but it is up to the authors to correct that. The manuscript should be more focalized to tissue engineering and to chitosan chemistry. Example reference 56 is outside this topic. The same is true for 61 (this material is intended for external application and not for bone). Reference 51 is also outside. Reference 47 is quite old, a new reference with similar chitosan (new materials) should be used. Reference 46 is also outside because, it belongs to edible films. Thus, outside the topic. Reference 39 is outside the topic (nanocellulose is not chitosan). Reference 34 is about PCL???? Chlorella vulgaris is also outside the topic (reference 32). R//We thank the reviewer for the suggestion. We decided to delete the following references from the last version to reduce the number of references and focus only on chitosan and Tissue engineering as suggested. 5. Dhandayuthapani, B.; Yoshida, Y.; Maekawa, T.; Kumar, D.S. Polymeric Scaffolds in Tissue Engineering Application : A Review. 2011, 2011, doi:10.1155/2011/290602. 32. Morais, F.P.; Simões, R.M.S.; Curto, J.M.R. Biopolymeric delivery systems for cosmetic applications using chlorella vulgaris algae and tea tree essential oil. Polymers (Basel). 2020, 12, 1–13, doi:10.3390/polym12112689. 34. Alipour, M.; Aghazadeh, M.; Akbarzadeh, A.; Vafajoo, Z.; Aghazadeh, Z.; Raeisdasteh Hokmabad, V. Towards osteogenic differentiation of human dental pulp stem cells on PCL-PEG-PCL/zeolite nanofibrous scaffolds. Artif. Cells, Nanomedicine Biotechnol. 2019, 47, 3431–3437, doi:10.1080/21691401.2019.1652627. 39. Asad, M.; Saba, N.; Asiri, A.M.; Jawaid, M.; Indarti, E. Preparation and characterization of nanocomposite fi lms from oil palm pulp nanocellulose / poly ( Vinyl alcohol ) by casting method. Carbohydr. Polym. 2018, 191, 103–111, doi:10.1016/j.carbpol.2018.03.015. 44. Shen, Z.; Kamdem, D.P. Development and characterization of biodegradable chitosan films containing two essential oils. Int. J. Biol. Macromol. 2015, 74, 289–296, doi:10.1016/j.ijbiomac.2014.11.046. 46. Villagómez-Zavala, D.L.; Gómez-Corona, C.; San Martín Martínez, E.; Pérez-Orozco, J.P.; Vernon-Carter, E.J.; Pedroza-Islas, R. Comparative study of the mechanical properties of edible films made from single and blended hydrophilic biopolymer matrices. Rev. Mex. Ing. Quim. 2008, 7, 263–273. 47. Hardwick, D.A. The mechanical properties of thin films: A review. Thin Solid Films 1987, 154, 109–124, doi:10.1016/0040-6090(87)90357-9. 51. Müller, K.; Zollfrank, C.; Schmid, M. Natural Polymers from Biomass Resources as Feedstocks for Thermoplastic Materials. Macromol. Mater. Eng. 2019, 304, 1–17, doi:10.1002/mame.201800760. 53. Martins, J.T.; Cerqueira, M.A.; Vicente, A.A. Influence of α-tocopherol on physicochemical properties of chitosan-based films. Food Hydrocoll. 2012, 27, 220–227, doi:10.1016/j.foodhyd.2011.06.011. 54. Mucha, M.; Pawlak, A. Thermal analysis of chitosan and its blends. Thermochim. Acta 2005, 427, 69–76, doi:10.1016/j.tca.2004.08.014. 56. Dashipour, A.; Razavilar, V.; Hosseini, H.; Shojaee-Aliabadi, S.; German, J.B.; Ghanati, K.; Khakpour, M.; Khaksar, R. Antioxidant and antimicrobial carboxymethyl cellulose films containing Zataria multiflora essential oil. Int. J. Biol. Macromol. 2015, 72, 606–613, doi:10.1016/j.ijbiomac.2014.09.006. 61. Vanegas, J.C.; Landinez, N.S.; Garzón-Alvarado, D.A. Generalidades de la interfase hueso-implante dental. Rev. Cuba. Investig. biomédicas 2009, 28, 130–146.
